# INTERACTIVE AGENT MODELING BY LEARNING TO PROBE

## ABSTRACT

The ability of modeling the other agents, such as understanding their intentions and skills, is essential to an agent's interactions with other agents. Conventional agent modeling relies on passive observation from demonstrations. In this work, we propose an interactive agent modeling scheme enabled by encouraging an agent to learn to probe. In particular, the probing agent (i.e., a learner) learns to interact with the environment and with a target agent (i.e., a demonstrator) to maximize the change in the observed behaviors of that agent. Through probing, rich behaviors can be observed and are used for enhancing the agent modeling to learn a more accurate mind model of the target agent. Our framework consists of two learning processes: i) imitation learning for an approximated agent model and ii) pure curiosity-driven reinforcement learning for an efficient probing policy to discover new behaviors that otherwise can not be observed. We have validated our approach in four different tasks. The experimental results suggest that the agent model learned by our approach i) generalizes better in novel scenarios than the ones learned by passive observation, random probing, and other curiosity-driven approaches do, and ii) can be used for enhancing performance in multiple applications including distilling optimal planning to a policy net, collaboration, and competition. A video demo is available at `https://www.dropbox.com/s/8mz6rd3349tso67/Probing_Demo.mov?dl=0`.

## 1 INTRODUCTION

An accurate understanding of other agents is essential to many multi-agent problems, such as collaboration, competition, and learning from an expert agent. Humans achieve this not only by passively observing others' behaviors, but also by actively probing others including interacting with them or changing the environment and conditions so that they can understand others' intentions, skills, and capabilities better. For instance, when working with a colleague for the first time, one may intentionally create diverse situations where the true intention and skill set of that colleague can be clearly revealed, which in turn helps improve the collaboration.

Inspired by this observation, in this work, we try to enable a probing agent (i.e., a learner) to automatically learn a good policy for probing in a way that helps it discover new behaviors of a target agent (i.e., a demonstrator) and thus learn a better model of the target agent that is generalizable to unseen environments or settings. Different from common task-oriented policy training, the learning of our probing policy is purely driven by the motivation of maximizing the knowledge about the target agent's model. We show a simple case in Figure 1 to illustrate this idea, where the learner and the demonstrator are initially located in the upper part and the lower part of the room respectively. The true policy of the demonstrator is trying to go to the upper part by finding the shortest path. However, since the room layout is fixed, the learner may overfit the only path observed from the demonstrator. By actively creating new gaps, the learner is able to discover various paths, which will greatly improve the accuracy of the approximated model of the demonstrator.

We consider the following setting for probing-based interactive agent modeling. In an environment, there are two general types of agents: i) a demonstrator who possesses certain skills for a single task or multiple tasks, and ii) a learner who has no prior knowledge of the environment and the demonstrator's skills. The purpose of the learner is to efficiently and thoroughly learn all of the demonstrator's skills and goals by not only passively watching the demonstrations but also actively interacting with the environment and/or the demonstrator. This learning process entails both imitation learning (IL) for modeling the demonstrator's skills and goals, and reinforcement learning (RL)

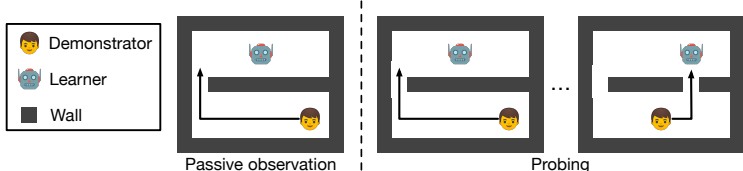

Figure 1: Illustration of our probing-based interactive agent modeling. Here, the demonstrator tries to go from the bottom-right corner to the upper part of the room. The passive learner (left) only observes one path in the fixed environment while the probing learner (right) removes a wall block to create a new gap so that the demonstrator will change its path accordingly.

for optimizing probing policy to diversify the task settings and the demonstrations to facilitate the imitation learning. Note that we assume that the demonstrator will always truthfully reveal its skills and intentions in any scenarios.

A key idea in our approach is to use task independent RL training purely driven by a curiosity reward. For this, we represent the demonstrator's mind by i) a latent vector to encode and track an agent's intention and belief, and ii) a policy conditioned on the latent vector and the agent's observed state for action prediction (i.e., the agent's skills). By introducing this latent vector, we are able to characterize an agent's policy by a low dimensional representation, and to reflect the change of policy by the change of this latent vector. Since the goal of probing policy is to cause the demonstrator to change its policy so that the learner may observe diverse demonstrations, it is natural to apply the change in the latent mind representation as the curiosity-driven incentive for the learner.

We evaluate our approach on four tasks in different domains (grid worlds and algorithmic problems). The experimental results indicate that our probing-based interactive agent modeling framework can: i) efficiently model the demonstrator's mind that is generalizable in unseen scenarios, and ii) can be applied to several applications including distilling optimal plans to a policy net by automatically diversifying task settings, and improving multi-agent collaboration as well as competition using the learned agent model.

## 2 RELATED WORK

In multi-agent reinforcement learning (MARL), agent modeling or opponent modeling plays an essential role as the ability of understanding other agent's goals and predicting their actions can greatly facilitate both collaborative and competitive purposes (Busoniu et al., 2008; Albrecht & Stone, 2018). Previous work has attempted to achieve this by task-oriented learning for maximizing a specified collaborative or competitive reward in the given tasks. For instance, inspired by game theory, there have been approaches aiming at finding Nash equilibira in multi-agent games, where agents' models are represented by their utilities (Littman, 1994; Hu & Wellman, 2003) and strategies (Claus & Boutilier, 1998; Tesauro, 2004; Powers & Shoham, 2005; Heinrich et al., 2015; Heinrich & Silver, 2016; Lanctot et al., 2017). Recently, some deep RL methods have incorporated simple agent modeling into Q-learning (Lowe et al., 2017) or policy updates (Foerster et al., 2018). Auxiliary tasks like explicitly predicting other agents' goals have also been applied to MARL (Mordatch & Abbeel, 2018). In previous work, the agents' incentive of modeling other agents comes from reaching a common goal or conflicting goals. In contrast, we never define a task-specific reward for the learner since the goal of our probing-based interactive agent modeling is not to reach a predefined goal but rather to learn a good mind model of the demonstrator which can be generalized to unseen settings and transferred to multi-agent tasks afterwards when a task-dependent reward is given.

Our work is greatly inspired by Theory of Mind (ToM) (Premack & Woodruff, 1978), which is a general and powerful framework to model an agent's mind and use the mind model to better explain or predict the agent's behaviors. Baker et al. (2009) has proposed a Bayesian formulation to incorporate an agent's desires, intentions, and belief about the world into the agent's policy in order to predict the agent's goals and actions via inverse planning. Rabinowitz et al. (2018) adopts a simpler mind representation (i.e., a latent vector) learned by a neural net (ToMnet). In our work, the learner is also trying to learn the policy of the demonstrator with a simple mind modeling. However, instead of only serving as a passive observer, we encourage the learner to probe so that it will learn to interact with the environment and with the demonstrator to quickly and continuously discover new behaviors of the demonstrator, which in turn helps learning a better mind model.

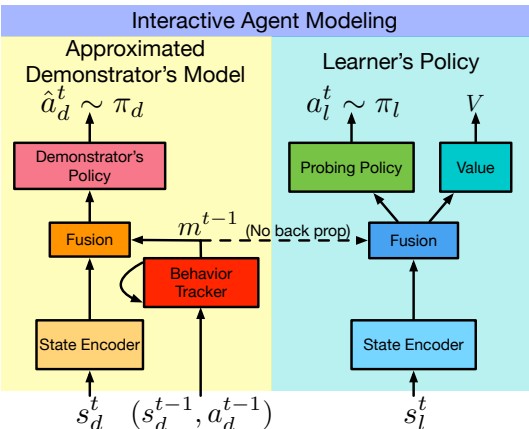

Figure 2: An overview of our model. Architecture details are in Appendix C. Note that the modules do not share weights, and the dashed line indicates that it is a feed forward only path (no back propagation through this path to update the mind model).

Our task-independent reward is related to the curiosity-driven reward applied to an RL agent for encouraging exploration (Strehl & Littman, 2008; Bellemare et al., 2016; Tang et al., 2017; Pathak et al., 2017). Our probing framework differs from this in two ways: i) instead of exploring the world states, we encourage the learner to discover new behaviors of the demonstrator to learn a better model of its mind; ii) the curiosity-driven rewards in previous work only serve as auxiliary rewards for achieving specific goals, whereas in our case, the sole motivation of our learner agent is from curiosity, and we demonstrate that this type of pure curiosity-driven learning can actually yield rich behaviors and general agent modeling.

There is certain similarity between active learning and our learning to prob mechanism. As Yang & Shafto (2017) shows, active learning is more effective than optimal teaching when the learner and teacher are not conceptually aligned, which is exactly the scenario in our problem setting (the learner does do not share any conceptual common ground with the demonstrator at the beginning). However, active learning typically addresses problems such as classification (Tong & Koller, 2001; Kapoor et al., 2007) by generating queries to an oracle to get additional ground-truth supervision. There are have been work on active imitation learning (Shon et al., 2007; nd Geoffrey J. Gordon & Bagnell, 2011; Judah et al., 2012) and active inverse reinforcement learning (Lopes et al., 2009) utilizing the similar concept, where the learner asks quires at certain states to a human oracle for guidance on what actions to take at those states. In contrast, our work goes beyond the scope of the existing work on active learning – we aim at training a learner agent to directly interact with the environment and with the target agent in order to automatically diversify the task settings and learn a better agent model without any task-dependent training objectives so that the learned agent models can be applied to improve the learner's performance in various applications.

Lastly, our task-independent learning objective can also be connected with meta-learning (Wang et al., 2016; Finn et al., 2017), which is to learn a meta strategy that can conduct efficient multi-task learning (Maclaurin et al., 2015; Duan et al., 2017; Hariharan & Girshick, 2017; Wichrowska et al., 2017; Yu et al., 2018; Baker et al., 2017) or adapt an agent's policy to its opponent's policy (Al-Shedivat et al., 2018) in a competitive setting. In this work, the purpose of our task-independent learning is to learn to probe a demonstrator for a better modeling of its mind, which is different from existing meta-learning approaches.

## 3 APPROACH

### 3.1 MODEL

We assume a Markov Decision Process (MDP) framework for both the demonstrator and the learner, where their behaviors at time $t$ are denoted by a pair of state and action $(s_d^t, a_d^t)$ and $(s_l^t, a_l^t)$ respectively. The history of their behaviors upon time $t$ is represented by trajectories $\Gamma_d^t = \{(s_d^\tau, a_d^\tau) : \tau = 1, \cdots, t\}$ and $\Gamma_l^t = \{(s_l^\tau, a_l^\tau) : \tau = 1, \cdots, t\}$ respectively.

Our interactive agent modeling framework is illustrated in Figure 2, which consists of two parts: i) learner's estimation of the demonstrator's model and ii) the learner's probing policy for a better understanding of the demonstrator's model.

To estimate the demonstrator's model, the learner maintains a behavior tracker, $\mathcal{M}(\cdot)$, to encode the observed trajectory of the demonstrator, which generates a latent vector, $m^t = \mathcal{M}(\Gamma_d^t)$. This latent vector can be viewed as a simplified representation of the demonstrator's mind upon time $t$, hence the learner may use it to characterize the demonstrator's policy, $\pi_d(a_d^t|s_d^t, m^{t-1})$, from which the learner may predict the demonstrator's future action $\hat{a}_d^t$. Note that for each demonstration, $m^t$ always starts from the same constant, $m^0 = \mathbf{0}$. This particular definition of the demonstrator's policy may also be connected with the option framework in hierarchical RL (Sutton et al., 1999), where the behavior tracker serves as a global policy to update the temporal abstraction $m^t$ and consequently changes the local policy $\pi_d$.

In this work, we require a learner to interact with the environment and/or with the demonstrator instead of passively watching the demonstrations. We enable this by learning a probing policy for the learner, $\pi_l(a_l^t|s_l^t, m^{t-1})$, where $m^{t-1}$ is from the current demonstration. The main purpose of the probing policy is to incite new behaviors of the demonstrator, thus we adopt a curiosity-driven reward to train this policy. Particularly, we define the reward function as

$$r^t = R(s_l^t, m^{t-1}, a_l^t, m^t) = ||m^t - m^{t-1}||^2, \qquad (1)$$

where $m^t$ is the successive output of the behavior tracker after observing $(s_d^t, a_d^t)$.

Finally, based on the probing policy, the learner can perform the probing as the rollout procedure outlined in Algorithm 1 (see Appendix A).

In summary, there are four key components in our probing-based interactive agent modeling:

- A behavior tracker $\mathcal{M}(\Gamma_d^t; \theta_M)$;
- The approximated demonstrator's policy $\pi_d(a_d^t|s_d^t, m^{t-1}; \theta_d)$;
- A probing policy for the learner $\pi_l(a_l^t|s_l^t, m^{t-1}; \theta_l)$;
- A value function for the probing policy $V(s_l^t, m^{t-1}; \theta_V)$.

Please refer to Appendix C for the details of the network architecture.

## 3.2 LEARNING

As discussed above, we have two main learning objectives corresponding to the two parts in our model respectively: i) minimizing imitation error (i.e., cross-entropy loss for action prediction) and ii) maximizing accumulated probing reward (i.e., probing policy optimization). Consequently, our approach includes an imitation learning process for recovering demonstrator's policy and a reinforcement learning process for optimizing the probing policy. These two processes are intertwined and influenced by each other: the IL process provides the behavior tracker guiding the probing policy while the RL process helps IL to observe more diverse behaviors from the demonstrator, thus enabling an interactive learning scheme. Algorithm 2 in Appendix A summarizes the overall learning approach, where $N$ is the total number of training iterations. The optimization details for the two learning processes are introduced as follows.

### 3.2.1 IMITATION LEARNING

For IL, we want to learn a good behavior tracker as well as the demonstrator's policy. For this, we minimize a cross-entropy loss for predicting demonstrator's actions:

$$\mathcal{L}(\theta_M, \theta_d) = \mathbb{E}\left[-\log \pi_d(a_d^t|s_d^t, m^{t-1})\right]. \qquad (2)$$

### 3.2.2 REINFORCEMENT LEARNING

The goal of RL is to train a good probing policy that will maximize the change of behavior and/or discover new behaviors of the demonstrator to facilitate the imitation learning. Based on the reward function defined in Eq. (1), this goal is equivalent to maximizing the accumulated reward, $J(\theta_l) = \mathbb{E}\left[\sum_{\tau=0}^{\infty} \gamma^\tau r^{t+\tau}\right]$, where $\gamma$ is the discounted factor.

For the policy optimization, we use Advantage Actor-Critic (A2C) (Mnih et al., 2016) to conduct on-policy training. The policy gradient is

$$\nabla_{\theta_l} J(\theta_l) = \nabla_{\theta_l} \left[ \log \pi_l(a_l^t | s_l^t, m^{t-1}; \theta_l) A(s_l^t, m^{t-1}, a_l^t) + \lambda \mathcal{H}(\pi_l(\cdot | s_l^t, m^{t-1}; \theta_l)) \right], \quad (3)$$

where $A(s_l^t, m^{t-1}, a_l^t)$ is the advantage estimation defined as $A(s_l^t, m^{t-1}, a_l^t) = \sum_{\tau=0}^{\infty} \gamma^\tau r^{t+\tau} - V(s_l^t, m^{t-1})$ and $\mathcal{H}(\cdot)$ is the entropy regularization weighted by the constant $\lambda = 0.01$ for encouraging exploration. The value function is updated by the following gradient:

$$\nabla_{\theta_V} \frac{1}{2} \left( \sum_{\tau=0}^{\infty} \gamma^\tau r^{t+\tau} - V(s_l^t, m^{t-1}; \theta_V) \right)^2. \quad (4)$$

Note that when we update the probing policy and the value function, the behavior tracker is fixed (i.e., no back propagation through the dashed path in Figure 2). Thus $\theta_M$ will only be updated by the IL loss in Eq. (2). This is to ensure that the change of $m^t$ is only caused by the change in policy or in behaviors, and not by the change of the parameters of the mind model, $\theta_M$.

## 4 EXPERIMENTS

To evaluate our approach, we introduce four tasks as shown in Figure 3, including three grid world tasks (passing through obstacles, maze navigation, construction) and an algorithmic problem (sorting).

In order to test the generalization ability of the learned agent model, we adopt a strict training procedure, where only one particular environment and task design is given during training. Specially, for grid world tasks, we fix the environment layout and/or item placement in each demonstration, whereas for the sorting task, we use the exact same input array throughout the training. At testing time, we randomize the task settings to an extent to create novel environments/inputs that have never been seen during training. We provide the specific settings in Section 4.1.

We implement rule-based policies for the demonstrator i) by searching the best plan from the initial state to the goal state for the grid world tasks or ii) by the bubble sort algorithm for sorting. When there is no possible path to reach the goal (e.g., blocked by the learner), the demonstrator will stop until a viable path appears.

### 4.1 TASKS

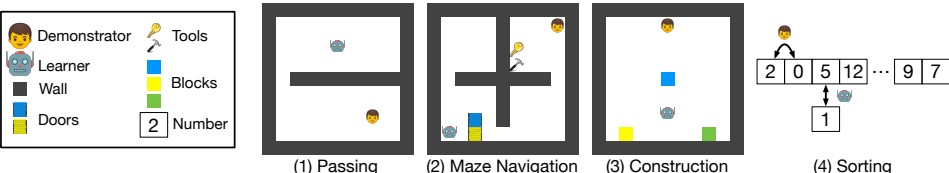

(1) Passing    (2) Maze Navigation    (3) Construction    (4) Sorting

Figure 3: Illustration of the evaluated tasks.

**Passing**. In this task modified from Baker et al. (2009), the demonstrator is initially located at the bottom-right corner and is trying to pass through the closest gap to get into the upper part of the room. The demonstrator can take 5 actions including moving in four directions and stopping, whereas the learner can move in four directions, stop, and also pickup or put down a wall block. The training environment is shown in Figure 3, where the gap is always located at the left end of the wall in the middle. In testing cases, we randomly place the location of the gap and the initial position of the demonstrator.

**Maze Navigation**. Inspired by similar tasks in recent literature (Andreas et al., 2017), we place a few door blocks and tools (a key and a hammer) in a four-room maze, where the key can be used to open the yellow door but has no effect on the blue door, which must be broken by the hammer. The demonstrator is trying to go from the top-right room to the top-left room. The two agents share the same action space including moving in four directions, picking up an item, and putting down an item. Also, they can only carry one item at a time. In the training setting, the initial positions of both agents and the door blocks are fixed as shown in Figure 3, whereas the tools may be randomly placed at only

a few locations. The rules in this environment are in fact fairly complex compared to other grid world tasks in previous work, where multiple sub-goals such as getting the tools, getting the door blocks, placing the door blocks, and walking through the doors are involved.

**Construction**. We adapt the stacking tasks in Shu et al. (2018) into a grid world, where the demonstrator has a latent goal invisible to the learner, which is to construct a new block by putting two blocks with a specific color combination together. Three items are present in a room and they are assigned with different colors randomly. In each episode, the demonstrator is randomly assigned with a goal (i.e., a pair of colors). It then seeks the needed blocks and puts one of them beside the other one. In order to predict the demonstrator's actions precisely, the learner must infer the correct goal first, which requires a sophisticated and dynamic agent modeling. Both agents share the same action space as in Maze Navigation. In training, there are no obstacles in the room. To increase the difficulty of goal inference, in testing scenarios, we randomly place a few wall blocks as obstacles around the colored blocks.

**Sorting**. Compared to a grid world, algorithmic problems are less visually informative and entail more abstract reasoning. For this, we design a sorting task where an array with certain length is given at the beginning. In our experiments, we set the length to be 10 and restrain the size of numbers in the array to be 4 bits (i.e., from 0 to 15). The demonstrator is able to perform a bubble sort algorithm to rearrange the input array in an ascending order. Its action at each step is to select a pair of numbers to swap. For every 5 steps done by the demonstrator, the learner can select a number and flipping one of the bit of that number. Both agents can choose to do nothing for a step. During training, we only provide one constant array so that the sorting always starts from the same initial array. This is a very challenging setting as only 10 out of 16 possible numbers are present in the training example and the fixed ordering may also easily cause overfitting. For testing, we generate random arrays as inputs.

For more details about the task settings, please refer to Appendix D.

## 4.2 GENERALIZATION IN UNSEEN TASK SETTINGS

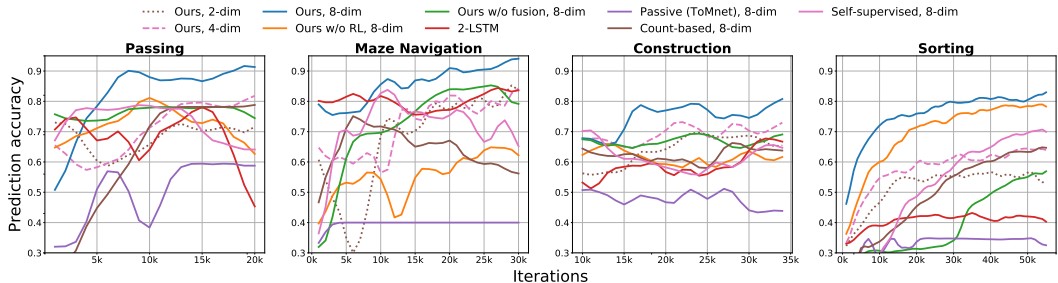

Figure 4: Action prediction accuracies in novel testing settings over numbers of training iterations.

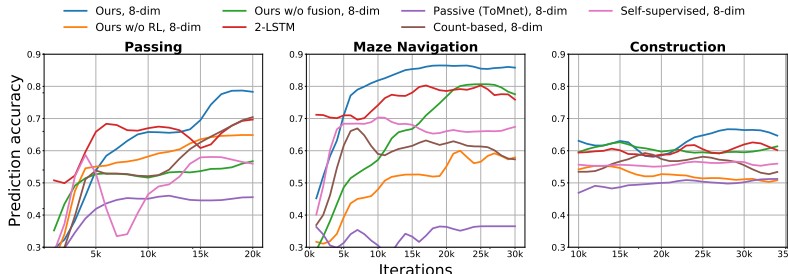

Figure 5: Action prediction accuracies in novel testing settings over numbers of training iterations with 10% random actions.

One of the main goals of learning to probe is to learn a good agent model that can be generalized to unseen scenarios. To evaluate how accurate our agent model is for approximating the true mind of the demonstrator, we may test the accuracy of predicting the demonstrator's actions using the learned $\pi_d$ and behavior tracker $\mathcal{M}(\cdot)$ in testing task settings unseen by the learner during training. A high

prediction accuracy in unseen settings will indicate good generalization of the learned agent model. To eliminate the effects from probing, we remove the learner from the environment during testing.

We compare our model with six baselines: i) ToMnet in Rabinowitz et al. (2018), which learns the demonstrator's model by only observing the given demonstrations without interactions, ii) our model without training the probing policy using RL (i.e., the leaner always takes random actions), iii), ours without the attention-based fusion (concatenating state feature and $m^{t-1}$ instead), iv) using two LSTMs for the estimated demonstrator's policy and the probing policy respectively (Figure 16), v) using count-based bonus as reward (Strehl & Littman, 2008; Bellemare et al., 2016; Tang et al., 2017), and vi) using cross-entropy loss for action prediction as reward (i.e., exploration by self-supervised prediction in Pathak et al. (2017)). To ensure fair comparison, the training settings and the testing settings are shared by all methods. We provide more details of the baselines in Appendix E.

Figure 4 shows the predication accuracy in testing settings of the three approaches based on the models from different training iterations. It is clear that with more iterations, our probing policy can greatly help increase the accuracy by discovering new behaviors, and consequently yields much higher testing accuracy than the baselines do. The results of "ours w/o fusion" and "2-LSTM" baselines further demonstrates the importance of our attention-based fusion layer and the use of a separate behavior tracker. By randomizing 10% of demonstrator's actions (Figure 5), we show that the probing policy can also handle stochastic and sub-optimal policies. It can be clearly seen from the results that the performance of the two baselines based on different curiosity rewards is clearly inferior to ours, which demonstrates the advantage of defining the behavioral change as the intrinsic reward for the purpose of agent modeling. We have also evaluated the robustness of our approach by showing the standard deviation from multiple runs as shown in Figure 9, which demonstrates a reasonably low variance across multiple runs.

We demonstrate the effect of dimensionality of the latent vector $m^t$ (i.e., the complexity of the agent model) in Figure 4. In simple environments, our approach still outperforms the baselines even when the dimensionality is decreased from 8 to 2 or 4. In more complex tasks like Sorting, a higher dimension is necessary for the agent modeling.

### 4.3 EVOLUTION OF LEARNED PROBING STRATEGY

As training progresses, we observe that our probing policy is able to progressively discover new behaviors through interactions that are adapted to the demonstrator's policy. For instance, in Maze Navigation, we find that the learner first learns to place one door, then gradually learns to place two doors at the appropriate moments to force the demonstrator to go back and forth to get the needed tools for opening the doors. Finally, the probing policy will even blocks the demonstrator for a while before it goes through the last door. Due to the space limit, we show this in the demo video.

We also provide more analysis and visualization of the probing behavior and the resulting latent vectors in Appendix B.2 and Appendix B.3.

### 4.4 EMERGENCE OF OBSTRUCTIVE BEHAVIORS FROM PROBING

Although we never explicitly set an adversarial goal for the learner, we do observe a natural emergence of obstructive behaviors caused by the probing, which can be quantitatively measured by the success rate of the demonstrator within a time limit as shown in Figure 6. This phenomenon is aligned with common sense that the optimal probing policy to discover new behaviors of the demonstrator should constantly force the demonstrator to change its plan, which will eventually delay the time when the demonstrator finishes the task. Because of the reward defined in Eq. (1), the probing policy learned from RL is also maximizing the accumulated behavioral change of the demonstrator just like the common sense. This further justifies our simple yet effective reward design.

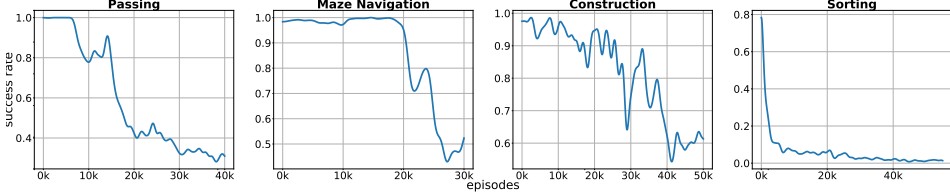

Figure 6: The average success rate of the demonstrator within the given time limit.

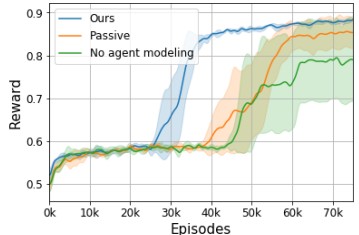 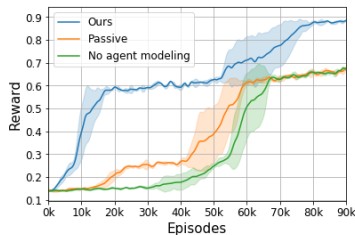

Figure 7: The learning curves of the collaborative task (reward is rescaled).

Figure 8: The learning curves of the competitive task (reward is rescaled).

## 4.5 Application 1: Distilling Optimal Plans to a Policy Net

Table 1: Success rates using the learned demonstrator's policy in unseen tasking settings. Unless specified, the evaluated policies were learned from demonstrations without random actions.

| Method | Passing | Maze Navigation | Construction | Sorting |
|---|---|---|---|---|
| Ours | **0.71** | **0.60** | **0.48** | **0.82** |
| Ours (from sub-optimal demo) | 0.68 | 0.51 | 0.47 | N/A |
| Ours w/o RL | 0.13 | 0.31 | 0.23 | 0.80 |
| Ours w/o fusion | 0.11 | 0.19 | 0.29 | 0.10 |
| Ours (2-LSTM) | 0.17 | 0.02 | 0.17 | 0 |
| Passive (ToMnet) | 0.11 | 0 | 0.12 | 0 |
| Count-based | 0.22 | 0 | 0.31 | 0.39 |
| Self-supervised | 0.23 | 0 | 0.36 | 0.56 |

Optimal planning sometimes requires a long computational time. For acceleration, it is common to distill the optimal plans to a policy net (Lazaric et al., 2010; Guo et al., 2014). However, the distilled policy net may not generalize well in new scenarios if the training settings are not diverse enough. Thus, the nature of our approach makes it suitable for improving the generalization without manually designing a large number of diverse settings.

For this, we evaluate the success rates when the learner directly uses $\pi_d$ (with an 8-dim latent vector) to perform tasks in testing settings without finetuning. The results summarized in Table 1 are consistent with the findings based on action predictions. We have also tested the success rate of the policy learned from sub-optimal demonstrations with 10% random actions. Its performance is comparable to the one learned from perfect demonstrations by our approach. It also outperforms the baselines trained from optimal demonstrations. We didn't test the randomized demonstrations for Sorting as it is unnecessary to randomize bubble sort algorithm. Note that since the learner is unaware of the goal in Construction, we let the learner take over the task after the first block has been picked up.

## 4.6 Application 2: Collaboration

To test whether the improved agent modeling by learning a probing policy can facilitate multi-agent collaboration, we modify the Construction task to be a collaborative task, where the learner is trying to help the demonstrator (fixed policy) to finish the task. For every step the demonstrator takes, the learner will get a -0.05 penalty. When the goal is reached, it will be given a reward of 1. This setting is difficult for training a collaborative policy since the demonstrator is capable of finishing the task by itself. In order to help finish the task faster, the learner must infer the true goal of the demonstrator quickly and shares part of the labor accordingly.

We fix the behavior tracker module, $\mathcal{M}(\cdot)$, trained from our interactive agent modeling and retrain the learner's policy $\pi_l$ using the task reward defined above. For comparison, we implement two baselines: i) retraining $\pi_l$ based on $\mathcal{M}$ learned from passive agent modeling (i.e., ToMnet) and ii) training a policy without agent modeling, i.e., $\pi_l(a_l^t | s_l^t, s_d^t)$.

Figure 7 demonstrates the learning curves, where the reward is rescaled so that the theoretical maximum reward from a perfect policy is 1. From the curves, we may see that the policy trained with our interactively learned agent model significantly outperforms both baselines.

### 4.7 APPLICATION 3: COMPETITION

Similar to Section 4.6, we design a competitive task based on the Construction task, where the learner gets a 0.05 reward for every step and a -1.0 penalty if the opponent achieves its goal. We adopt the same training procedure as in Section 4.6, and also rescale the reward according to the maximum reward. As Figure 8 shows, the mind model learned by our approach improves the learning efficiency and the converged reward by a large margin.

## 5 CONCLUSIONS

In this work, we have proposed a novel agent modeling approach, i.e., probing-based interactive agent modeling. The core idea is to learn a probing policy using only a curiosity-driven reward, which is able to discover new behaviors of the target agent. We achieve this by incorporating two learning processes (IL and RL) together. We are able to validate our approach in four distinct tasks. The results show that by learning a probing policy, the learner in our approach can build a more accurate agent model of the demonstrator. Thanks to this interactively learned agent model, the learner is able to i) approximate the demonstrator's policy more accurately in unseen settings compared to passive agent modeling, ii) efficiently learn a good collaborative policy to help the demonstrator, and iii) develop an adversarial policy to compete with the demonstrator.

In the future, we can extend this framework to simultaneous agent modeling and world modeling with a more complex mind representation.

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

## A   PSEUDO CODE OF OUR ALGORITHMS

---

**Algorithm 1** Rollout($T_{\max}$)

---

**Input:** Maximum steps $T_{\max}$
**Output:** Episode length $T$, trajectories $\Gamma_d^T$ and $\Gamma_l^T$, and the latent vector sequence $M$
 1: Initialize the environment
 2: $\Gamma_d^0 \leftarrow \emptyset, \Gamma_l^0 \leftarrow \emptyset, M \leftarrow \emptyset, m^0 \leftarrow \mathbf{0}, t \leftarrow 0$
 3: **repeat**
 4:     $t \leftarrow t + 1$
 5:     Observe $s_d^t$ and $a_d^t$ from the demonstrator
 6:     Observe $s_l^t$ from the environment
 7:     Sample and execute the learner's action $a_l^t \sim \pi_l(s_l^t, m^{t-1})$
 8:     $\Gamma_d^t \leftarrow \Gamma_d^{t-1} \cup \{(s_d^t, a_d^t)\}, \Gamma_l^t \leftarrow \Gamma_l^{t-1} \cup \{(s_l^t, a_l^t)\}$
 9:     $m^t \leftarrow \mathcal{M}(\Gamma_d^t), M \leftarrow M \cup \{m^t\}$
10: **until** $t = T_{\max}$ or the task is finished
11: $T \leftarrow t$

---

**Algorithm 2** Learning Algorithm

---

 1: Initialize parameters $\Theta = \langle \theta_M, \theta_b, \theta_l, \theta_V \rangle$
 2: Set $T_{\max}$ (the maximum steps in an episode) and $N$ (the number of training iterations)
 3: $i \leftarrow 1$
 4: **repeat**
 5:     $T, \Gamma_d^T, \Gamma_l^T, M \leftarrow$ Rollout($T_{\max}$)
 6:     IL: Update $\theta_M$ and $\theta_d$ based on Eq. (2) using $\Gamma_d^T$
 7:     RL: Update $\theta_l$ and $\theta_V$ based on Eq. (3 and Eq. (4) respectively using $\Gamma_l^T$, and $M$
 8:     $i \leftarrow i + 1$
 9: **until** $i = N$

---

## B   MORE RESULTS

### B.1   ROBUSTNESS EVALUATION

We show the mean and standard deviation of the prediction accuracy from 5 runs by our full model in Figure 9 for Maze Navigation to validate the robustness of our approach.

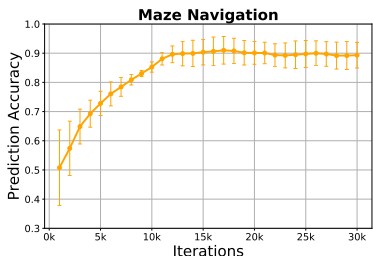

Figure 9: Mean and standard deviation of multiple runs in Maze Navigation.

### B.2   VISUALIZATION OF LATENT VECTORS

Figure 10 visualizes the latent vectors obtained from demonstrations with probing and without probing, where the latent vectors were computed by the same behavior tracker in both cases. This provides empirical evidences that by finding new latent vectors, we are able to discover new demonstrations with probing.

### B.3   VISUALIZATION OF THE CHANGE IN $m^t$ AND THE CHANGE IN POLICY

To show that the change in $m^t$ indeed indicates the change in policy, we compute the correlation of $||m^t - m^{t-1}||^2$ and $KL(\pi_d(\cdot|s^{t+1}, m^t)||\pi_d(\cdot|s^{t+1}, m^{t-1}))$ (i.e., how different the policy conditioned

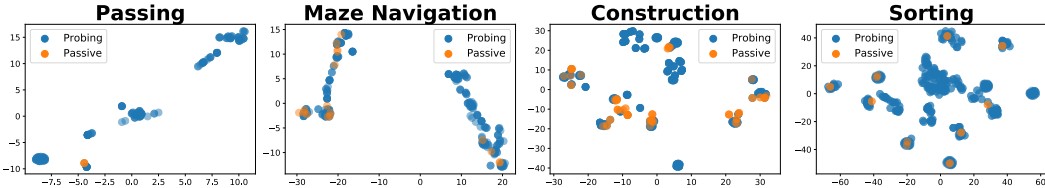

Figure 10: t-SNE embedding of $m^t$.

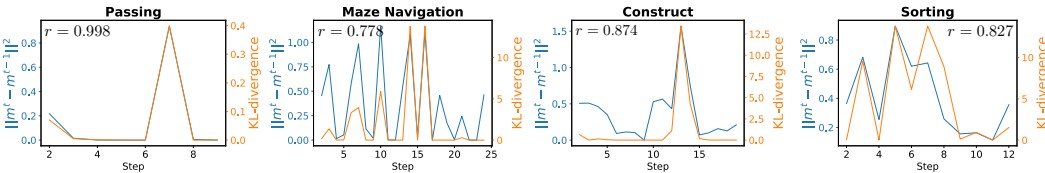

Figure 11: Correlation between the change in $m^t$ and the change in policy in testing settings ($r$ is Pearson correlation coefficient).

on the new latent vector $m^t$ is compared to the one with the old latent vector $m^{t-1}$). Figure 11 demonstrates the correlation between the change in $m^t$ and the corresponding change in policy in testing settings. The high correlation validates our hypothesis that the distance between consecutive latent vectors $m^t$ and $m^{t-1}$ reflects the policy change of the demonstrator.

## C  NETWORK ARCHITECTURE OF OUR MODEL

**State Encoder**. The input of the state encoder is a multi-channel tensor. For the grid world case, the input dimension is $11 \times 11 \times (N_{\text{blocks}} + 1)$, where $11 \times 11$ is the size of the grid world, $N_{\text{items}}$ is the number of types of blocks, and the additional channel is to show the position of the corresponding agent, i.e., the position of the demonstrator for $s_d^t$ or the position of the learner for $s_l^t$. The other agent is treated as an obstacle and its position is encoded into the channel corresponded to the wall block. In the case of Sorting task, the input dimension is $10 \times 1 \times 4$, representing 10 numbers in an array where the size of each number is 4 bits. The state encoder has one convolutional layer which consists of 32 filters with kernel size of $1 \times 1$ and stride of 1.

**Behavior Tracker**. Assuming the action space is $A$, we combine the state input and the action input by augmenting the state input with $A$ channels, each of which corresponds to an action. We set the channel of the observed action to be all ones and set the remaining $A - 1$ channels to be zeros. This combined state and action input is then fed into a convolutional layer with 32 filters (the kernel size is $1 \times 1$ and the stride is 1). The output is flatten into a vector and passed through two fully connected (FC) layers (all have 128 dimensions). The resulting 128-dim vector serves as the input of an LSTM with 128 hidden units. Finally, an FC layer takes in the hidden state from the LSTM and outputs the latent vector $m^t$ as the mind representation.

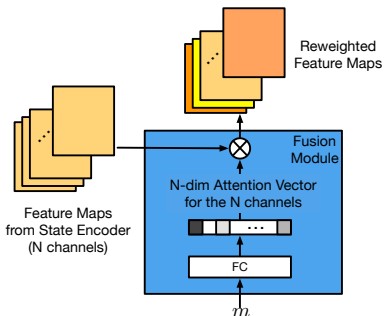

Figure 12: The attention-based fusion module.

**Fusion**. As shown in Figure 12, we design our fusion module using an attention based mechanism similar to the one introduced by Chaplot et al. (2017), where the latent vector $m^{t-1}$ is fed into an FC layer outputting an $N$-dim attention vector (each element is from 0 to 1) corresponding to the $N$ feature maps from the state encoder (here $N = 32$). Formally, we have an attention vector $h = \sigma(m^{t-1}) \in \mathbb{R}^{32}$, where $\sigma(\cdot)$ is an FC layer with sigmoid activation. $h$ is spatially expanded to a $H \times W \times 32$ tensor, $\boldsymbol{H}(h) \in \mathbb{R}^{H \times W \times 32}$, where the elements in $k$-th channel correspond to the $k$-th element in $h$. We then reweight each feature maps using the attention vector, which becomes the fusion output. I.e., $f(\phi(s^t), m^{t-1}) = \phi(s^t) \odot \boldsymbol{H}(\sigma(m^{t-1}))$, where $\phi(s^t)$ are the feature maps from the state encoder, $f(\cdot)$ is the fusion layer, and $\odot$ is element-wise product.

**Policy**. The input of this module is the flattened output from the fusion module, and is fed to an LSTM with 128 hidden units followed by an FC layer with softmax activation. The resulting output is an action distribution representing the policy (either $\pi_d$ or $\pi_l$). For Sorting task, we slightly modify the output to fit the problem. We decompose the demonstrator's policy as $\pi_d(a_d^t|s_d^t, m^{t-1}) = \pi_d^{(1)}(a_d^{t,1}|s_d^t, m^{t-1})\pi_d^{(2)}(a_d^{t,2}|s_d^t, m^{t-1})$, where $a_d^{t,1}$ and $a_d^{t,2}$ are the indices of the numbers the demonstrator chooses to swap. For the learner's policy, it is decomposed as $\pi_l(a_l^t|s_l^t, m^{t-1}) = \pi_l^{\text{id}}(a_l^{t,1}|s_l^t, m^{t-1})\pi_l^{\text{bit}}(a_l^{t,2}|s_l^t, m^{t-1})$ instead, where $a_l^{t,1}$ indicates the number that the learner selects to change and $a_l^{t,2}$ is the bit of that number that needs to be flipped. When $a_d^{t,1}$ or $a_l^{t,1}$ is larger than the length of the array, it means that the demonstrator or the learner is choosing to do nothing respectively.

**Value**. We also have a value net designed for training the learner's policy using A2C (i.e., $V(s_l^t, m^{t-1})$), which takes in the hidden state from the LSTM in the learner's policy module and outputs a scalar value after an FC layer.

The network is trained with RMSProp (Tieleman & Hinto, 2012) using a learning rate of 0.001. During training, $\epsilon$-greedy is applied to the rollout, where the $\epsilon$ gradually decreases from 0.1 to 0.01.

## D   TASK SETTINGS

We assume full observations of the world state for both agents in all tasks but the internal state of an agent (e.g., goals) is unobservable to another agent. The discounted factor is set to be $\Gamma = 0.95$.

For the demonstrator in the grid world tasks, we implemented search based path planning and used simple heuristics to perform branch and bound for acceleration. In particular, the state for the search algorithm in Passing is the map status, whereas the state in Maze Navigation and Construction is the combination of map status and the demonstrator's inventory.

### D.1   PASSING

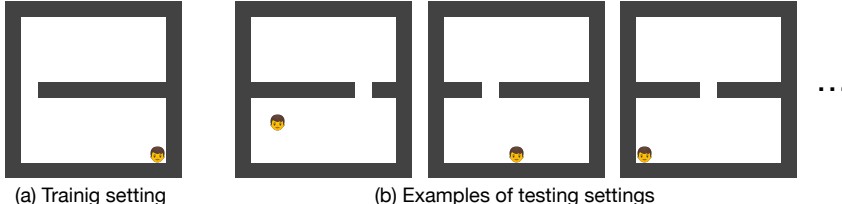

(a) Trainig setting          (b) Examples of testing settings

Figure 13: The training setting and examples of testing settings for Passing.

Figure 13 shows the training setting where the locations of the gap and the staring point of the demonstrator are fixed, and the examples of testing settings where the placement of the gap and the initial position of the demonstrator is randomized. We terminate a training episode if the demonstrator has not passed the obstacle after 15 steps.

### D.2   MAZE NAVIGATION

The training setting in Maze Navigation is designed as shown in Figure 14a, where the placement of the tools is restrained in the purple region, and the positions of the demonstrator's starting point and the doors are fixed. For testing, we randomly put one or two doors to fill the gaps; the demonstrator

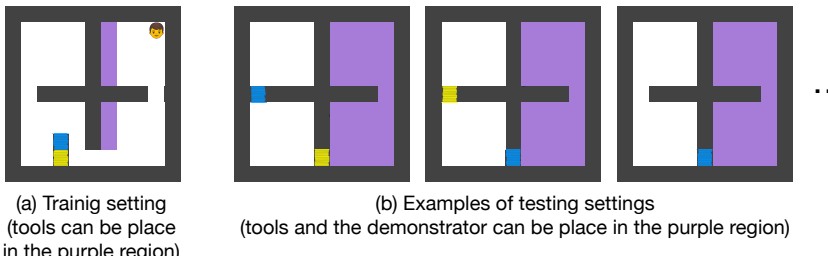

(a) Trainig setting
(tools can be place
in the purple region)

(b) Examples of testing settings
(tools and the demonstrator can be place in the purple region)

Figure 14: The training setting and examples of testing settings for Maze Navigation.

and the tools can be randomly placed in the purple region. The demonstrator is always guaranteed to be able to find a path from its starting point to the destination (the top-left room). A training episode has a time limit of 60 steps.

### D.3 CONSTRUCTION

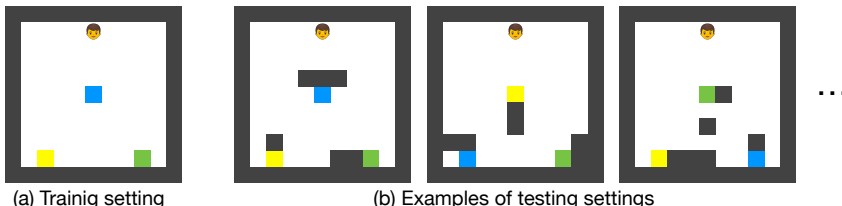

(a) Trainig setting

(b) Examples of testing settings

Figure 15: The training setting and examples of testing settings for Construction.

The room layout in training setting is fixed and shown in Figure 15a. In testing settings, we randomly put six wall blocks around the three colored blocks to create obstacles. Figure 15b displays a few examples of testing scenarios. Note that in both training and testing, we allow randomized coloring as long as the goal can be achieved.

For each episode, we assign a random goal (a pair of colors) for the demonstrator. The maximum episode length is 30 steps during training.

### D.4 SORTING

---
**Algorithm 3** Modified Bubble Sort

---
**Input:** Initial array $X = [x_1, x_2, \cdots, x_n]$, where $n$ is the length.
**Output:** Sorted array
1: Last position $i \leftarrow 0$
2: Steps $t \leftarrow 0$
3: **while** $X$ is not in an ascending order **do**
4:      $c \leftarrow 0$
5:      **while** $c < n - 1$ **do**
6:          **if** $x_i > x_{i+1}$ **then**
7:              Swap $x_i$ and $x_{i+1}$, i.e., the demonstrator's $t$-th action is $(i, i+1)$
8:              $t \leftarrow t + 1$
9:              break
10:          **end if**
11:          $c \leftarrow c + 1$
12:          $i \leftarrow (i+1)\%(n-1)$
13:      **end while**
14: **end while**

---

In training, there is only one sequence as the initial state, i.e., [2, 0, 5, 12, 14, 10, 3, 11, 9, 7]. The testing settings include 100 randomly generated initial sequences. Training episodes have a 30-step time limit.

Since the learner may change certain numbers during the process of sorting, the original bubble sort may fail to finish the sorting successfully since it will not look back at the sorted part of the array. To address this, we modify the original bubble sort algorithm so that it will continue to sort the sequence until it is in an ascending order. Algorithm 3 outlines how the demonstrator swaps the numbers.

## E    DETAILS OF BASELINES

### E.1    REWARD FUNCTIONS IN BASELINES

We define the reward functions used for baselines, count-based reward and self-supervised prediction here.

**Count-based reward**:

$$r^t = R(s_d^t) = \frac{\beta}{\sqrt{N(s_d^t)}},$$

(5)

where $\beta$ is a constant (we set $\beta = 1$, which gives the best results in our experiments), and $N(s_d^t)$ is the counts of state visitation. In our experiments, the counting can be efficiently implemented by hashing. This reward encourages the learner to push the demonstrator to new states in order to incite new demonstrations.

**Self-supervised prediction**:

$$r^t = R(s_d^t, m^{t-1}, a_d^t) = -\log \pi_d(a_d^t|s_d^t, m^{t-1}),$$

(6)

where $a_d^t$ is the ground-truth action from the demonstrator. This reward essentially measures action prediction loss of the estimated demonstrator's policy, which is designed to encourage the learner to find new scenarios where the previously learned demonstrator's policy becomes less accurate.

### E.2    NETWORK ARCHITECTURE OF THE 2-LSTM BASELINE

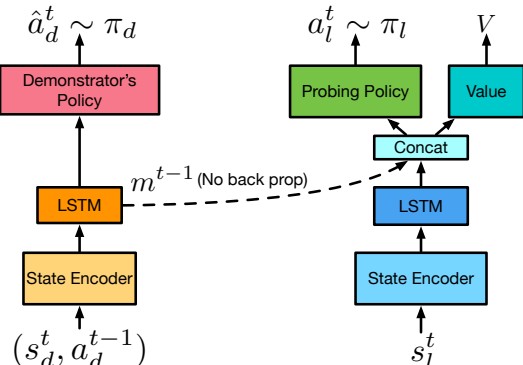

Figure 16: The network architecture of the 2-LSTM baseline.

Figure 16 illustrates the network architecture of the 2-LSTM baseline, where two LSTMs all have 128 hidden units.

