# OpenReview forum: "Interactive Agent Modeling by Learning to Probe"
_ICLR.cc/2019/Conference_

### Official Review · AnonReviewer2 · 2018-11-01
**Learning to model other static agents in the environment. Compelling idea but limited evaluation.**

**Rating:** 6
**Confidence:** 4

**Review:**

The submission proposes a new method for agent design to learn about the behaviour of other fixed agents inhabiting the same environment. The method builds on imitation learning (behavioural cloning) to model the agent’s behaviour and reinforcement learning to learn a probing policy to more broadly explore different target agent behaviours. Overall, the approach falls into the field of intrinsic motivation / curiosity-like reward generation procedures but with respect to target agent behaviour instead of the agent’s environment. While learning to model the target agent’s inner state, the RL reward is generated based on the difference of the target agent’s inner state between consecutive time steps.

The approach is evaluated against a small set of baselines in various toy grid-world scenarios and a sorting task and overall performs commensurate or better than the investigated baselines. Given its limitation to small and low-dimensional environments, it cannot be said how well the approach will scale with respect to these factors and the resulting, more complex agent behaviours. It would be highly beneficial to evaluate these aspects. Furthermore, it would be beneficial to provide more information about the baselines; in particular the type of count-based exploration. For the generated figures, it would be beneficial to include standard deviation and mean over multiple runs to not only evaluate performance but also robustness.

Overall, while the agent behaviour modelling focused on a type of inner state (based on past trajectories) provides benefits in the evaluated examples, it is unsure how well the approach scales to more complex domains based on strong similarity and simplicity of the tested toy scenarios (evaluation on sorting problems is an interesting step towards to address this shortcoming). One additional aspect pointing towards the necessity of further evaluation is the strong dependence of performance on the dimensionality of the latent, internal state (Fig.4).

Minor issues:
- Reward formulations for the baselines as part of the appendix.
- Same scale for the y-axes across figures

---

> ### Author Response · Authors · 2018-11-25
> **Author response**
>
> Thank you for your reviews and comments. We respond to your questions as follows.
>
> 1. Scalability?
> While we agree that the tasks in this paper are not real world problems, we think, as a first step towards this direction, the evaluations in this paper have provided some promising proof-of-concept results. Applying the approach to more realistic and more complex tasks could be a good future research direction.
>
> 2. It would be beneficial to provide more information about the baselines
> We have added details of baselines including their reward functions in Appendix E.
>
> 3. For the generated figures, it would be beneficial to include standard deviation and mean over multiple runs
> We show the standard deviation of multiple runs in Figure 7,8,9 in the revision. We have done our best to evaluate the robustness given the limited time and will continue to improve the evaluation.
>
> 4. The strong dependence of performance on the dimensionality of the latent, internal state (Fig.4).
> The network architecture design is not the focus of our paper. Generally speaking, a higher dimensionality of the latent vector provides a more powerful network to model agents. However, as we show in Figure 4, with probing, the network with lower dimensionality can even outperform the baselines trained with latent vectors that have higher dimensions. And with the same architecture, probing clearly provides a significant improvement.
>
> 5. Minor issues.
> Thanks for pointing out these issues. We have fixed them in the revision.

---

> > ### Comment · AnonReviewer2 · 2018-12-04
> > **Rev Response**
> >
> > Thank you very much for addressing some of the crucial points mentioned in the review. In particular, details on the baseline are appreciated as it represented a significant shortcoming with respect to reproducibility as well as the inclusion of std. deviation on at least some of the experiments.
> > While it is natural that the main criticism (scale) of the original review is the most time involving of the relevant points, it would have been beneficial to write about possible directions in which the issue can be addressed in the next steps.
> > While not all required aspects (with respect to this review) to fully ensure justification for a change of rating are addressed, the extension with respect to all reviews (including other reviewers) and more detailed evaluation and discussion enable me to increase the rating when taking into account overall improvements.
> > This being said, with and without adaptation all reviews seem very much aligned.

---

### Official Review · AnonReviewer3 · 2018-11-05
**"Interactive Agent Modeling by Learning to Probe" provides an interesting approach to improve imitation learning**

**Rating:** 6
**Confidence:** 4

**Review:**

This paper presents a method for interactive agent modeling that involves learning to model a demonstrator agent not only through passively viewing the demonstrator agent, but also through interactions from a learner agent that learns to probe the environment of the demonstrator agent so as to maximally change the behavior of the demonstrator agent. The approximated demonstrator agent is trained through standard imitation learning techniques and the learning or probing agent is trained using reinforcement learning. The mind of the demonstrating agent is modeled as a latent space representation from a neural net. This latent space representation is used as the reinforcement learning signal for the learner (probing) agent similar to the curiosity driven techniques where larger changes in the representation of mind are sought out since they should lead to larger differences in demonstrator agent behavior. The authors test this in several gridworld environments as well as a sorting task and show that their method achieves superior performance and generalizes better to unseen states and task variations compared to several baseline methods.

General comments, in no particular order:

1. The authors should provide more details on how the hand-crafted demonstrator agents were made. I assume something similar to an a* algorithm was probably used for the passing task, but what about the maze navigation task?

2. The demonstrated tasks are (gridworld and algorithmic) which are very simple RL taks with low-dimensional (non-visual) state-spaces.  It's unclear how this would scale to more complex tasks with higher-dimensional state spaces such as Atari, Starcraft II or if this would work with tasks with continuous state and action spaces such as mujoco.

3. The core premise behind training the learner agent with RL is using a curiosity driven approach to train a probing policy to incite new demonstrator behaviors by maximizing the differences between the latent vectors of the behavior trackers at different time steps. Because the latent vector is modeled as a non-linear function, distances between latent vector representations do not necessarily correspond to similar distances between behavior policies (for example, KL distances between two policy distributions). Since this is for ILCR, I think the authors should have taken a deeper dive into examining those latent representations and potentially visualizing those distances and how they correspond to different policy behaviors.

4. The biggest flaw that I see in this method is the practicality of it's use. This method relies on the ability to obtain or gain access to a demonstration agent to learn from. In very simple tasks, such as the one presented here, the authors were able to hard-code their own demonstration agent. However, in harder tasks, this will not be feasible. If you are able to obtain or code your own agent, then you've already solved the task and there is no need to do any sort of imitation learning in the first place.  In reality, for sufficiently difficult tasks, a human would be the demonstration agent (as is done in most robotics tasks). In practice, imitation learning from a human works well since the learning can be done offline (i.e., post-hoc after a set of demonstrations are collected from the human). However, this task requires the learning to be interactive and thus the demonstrator needs to be present during the learning.  Interactively learning from a human becomes a problem if the learning takes tens of thousands of episodes of training since a human cannot reasonably be expected to be present for that amount of time. Thus, the question is 1) how well will this method work with a human acting as the demonstrator? and 2) how can this method work if you are not able to have access to a demonstrator long periods of time (or even at all)?

5. My previous comment relates mainly to the application of improved imitation learning. However, I do think this is still very useful in the context of multi-agent reinforcement learning for collaborative and competitive tasks (sections 4.6 and 4.7). I think this method demonstrates a method for improved collaborative and/or competitive performances given the fact that you already have a single agent with a learned policy.

Overall, I think the paper presents a really nice idea of how to improve modeling of agents. essentially, a learner agent learns how to probe a demonstrator agent to provide more information about what's being demonstrated and prevent over-fitting to a set of fixed demonstrations.   This work sounds novel to me from a reinforcement learning perspective, however, I'm not well versed on theory of mind research.

---

> ### Author Response · Authors · 2018-11-25
> **Author response**
>
> Thank you for your detailed reviews. Here are our responses to your questions and concerns.
>
> 1. The authors should provide more details on how the hand-crafted demonstrator agents were made.
> We have added more details, and plan to release the code. We indeed implemented search algorithm with simple heuristics for acceleration for all grid-world tasks. In Maze Navigation, the state space is extended to the combination of map status and the agent's inventory. By this definition of states, an efficiency search can still be achieved.
>
> 2. Scalability?
> We focus on simpler domains to provide proof-of-concept results as the first step on this direction. We are definitely interested in studying how our approach can be applied to more complex tasks as future work.
>
> 3. A deeper dive into examining those latent representations and potentially visualizing those distances and how they correspond to different policy behaviors.
> Thanks for the suggestion. We have included a more detailed analysis with new visualizations in the updated paper. i) We visualize the latent vectors obtained from demonstrations with probing and without probing. It indeed shows that with probing, we are able to find new behaviors that correspond to the new latent vectors. ii) We also show the correlation between the distance of two consecutive latent vectors m^{t-1} and m^t and, the KL divergence between the two corresponding policies KL(\pi(a|s^{t+1},m^t) || \pi(a|s^{t+1},m^{t-1})), i.e., how different the policy would have been if m^t didn’t change. The correlation is significant, and thus validates the idea.
>
> 4. 1) how well will this method work with a human acting as the demonstrator? and 2) how can this method work if you are not able to have access to a demonstrator long periods of time (or even at all)?
> We focus on improving modeling machine agents, and applying the improved agent models for multi-agent tasks. The current form of our approach is not designed for learning from human demonstrations. However, there are ways to modify our approach towards that direction: i) learning probing policy with model-based RL; ii) incorporating inductive bias from humans (e.g., the learner knows a specific set of possible goals of the demonstrator and probes the demonstrator to test which goal it has). This seems to be a good direction for future work, but we also think that the current research has provided promising results in simpler domains, and hopefully incites more research where human demonstrators are also involved by introducing this problem to the community.
>
> 5. I think this method demonstrates a method for improved collaborative and/or competitive performances given the fact that you already have a single agent with a learned policy.
> Yes, in our experiment, we do assume that the opponent has a learned policy which is unknown to us. We think that this is a quite general setting where multiple machine agents are interacting with each other but do not know each other’s true policies and intentions.

---

> > ### Comment · AnonReviewer3 · 2018-12-06
> > **reviewer response**
> >
> > I think the authors have done a good job at addressing my previous comments/concerns, including performing additional analysis in examining the latent representations. Figure 11 shows convincing evidence that the difference in latent vectors correspond well to differences in policy distributions. Figure 10, however, is not very well explained and is unclear what it is showing or what it's significance is. Why is there only one orange dot in the passing environment? What does each dot represent? What are the significance of the clusters that form? It would be interesting to tie these clusters back to the behaviors to see if they are qualitatively different.

---

> > > ### Author Response · Authors · 2018-12-07
> > > **Author response**
> > >
> > > Thank you very much for your reply. In Figure 10, each dot represents a specific latent vector m^t. Basically we plot all m^t that appeared in multiple episodes. Since we have qualitatively shown in the demo video that probing can incite new behaviors, the fact that we do observe new latent vectors m^t with probing in Figure 10 supports our idea that inciting new behaviors can be achieved by finding new latent vectors m^t. There are actually more than one orange dots in Passing -- they just have the same coordinates in the t-SNE embedding. This is expected because without probing, all the demonstrations are exactly the same. We will clarify the meaning of this figure.

---

### Official Review · AnonReviewer1 · 2018-11-08
**Nice work, more details and some references to previous work needed**

**Rating:** 6
**Confidence:** 3

**Review:**

The authors consider the scenario of two agents, a demonstrator acting in an environment to achieve a goal, and a learner, which can also interact with the environment, but whose goal is to learn the demonstrator’s policy by carrying out actions eliciting strong changes in the demonstrator’s trajectory. The former is implemented as imitation learning, i.e. policy learning, the latter as curiosity driven RL.

The authors are encouraged to review some of the related literature on optimal teaching, which also has developed a rich set of approaches to agent modeling, e.g. the work by Patrick Shafto. It may also be relevant to think about the relationship to active learning in IRL.

I am not sure whether I would be able to implement and reproduce the presented work on the basis of the current manuscript including the appendix. It would be very helpful for the community to be able to do so. E.g., details on the the training of the demonstrators, their reward functions, and the behavior tracker. Particularly the "fusion" module remains extremely unclear.

Overall, this is a nice paper, despite the fact that the example domains and problems considered are engineered strongly to allow for the proposed algorithm to be useful. Particularly for the claim of generalization to different environments, the details are all in the engineering of the particular grid world tasks, how they relate to each other and the sate representation used for the demonstrator s_d. I am not sure why it was submitted to ICLR and not the Annual Meeting of the Cognitive Science Society, though.

Minor points:
“differs from this in two folds”
“by generate queries”

---

> ### Author Response · Authors · 2018-11-25
> **Author response**
>
> Thank you for your comments and suggestions. Please see our responses below.
>
> 1. Related work
> Thanks for pointing out this. We have added discussion about the optimal teaching and active IRL.
>
> 2. More implementation details
> We have provided more details in the revision and plan to release our code. Regarding your questions: i) demonstrators policies are implemented by search algorithms; ii) the behavior tracker is an LSTM with 128 hidden units; iii) fusion module produces a 32-dim attention vector corresponding to 32 feature maps from the state encoder, and each element of that vector is used to reweight one of the feature map in order to reshape the state feature.
>
> 3. I am not sure why it was submitted to ICLR and not the Annual Meeting of the Cognitive Science Society
> We think this is appropriate for ICLR as we propose a novel deep RL approach to improve representation learning for agent modeling. Having said that, it could be an interesting future work to study how humans perform probing in the perspective of cognitive science.
>
> 4. Typos
> Thanks for point out the typos. We have fixed them in the revision.

---

### Official Review · AnonReviewer4 · 2018-11-11

**Rating:** 6
**Confidence:** 4

**Review:**

1) Summary
This paper proposes a method for learning an agent by interacting and probing an expert agents behavior. This method is composed of a policy that learns to imitate an expert’s action, and a policy that challenges the expert in order to get it to take multiple possible routes to solve a task. The two policies share a “behavior tracker” that models the expert’s behavior, and communicates it to both policies being learned. The probing policy is optimized using a curiosity-driven reward in order to get the expert take trajectories the probing policy has not seen before. In experiments, the authors perform experiments to show how the learned agent can generalize to unseen configurations in the corresponding environments in which the agents were trained, and also use the proposed technique in a sorting task in which the method generalizes to longer arrays to be sorted.


2) Pros:
+ Neat idea for exploring an experts behavior by changing the environment surrounding it (probing it).
+ Cool experiments for applicability.
+ Well written paper and easy to understand.

3 Comments:
- Equation 1 typo?:
To my understanding, in curiosity driven exploration, the exploration is driven based on how well the next state can be predicted by the agent. In equation 1, different time steps are being compared, m^t and m^{t-1}, but the comparison should be between the predicted time step t and real time step t. Can the authors clarify why different time steps are compared in the equation?

- Baseline missing: Random actions from expert
A simple baseline to compare against could be to simply force the expert to take a few random actions during its trajectory and let the imitator learn from these. Comparing against this baseline could serve as evidence that we need to actually learn the probing agent to acquire a more optimal policy.

- Baseline missing: Simple RNN policies that communicate hidden states.
Another baseline could be to simply model the imitator and probing policies as RNNs and let them communicate with each other via the hidden states. While optimizing the curiosity reward the hidden states could be used as well. If successful, this baseline can show that we actually need to model the “behavior” with a separate network.

- Ablation study for the importance of fusion:
The authors have a “fusion” layer within the imitator and probing policies. An ablation study showing that this layer is actually necessary is missing from the paper.

- Generalizability argument
The authors claim that they show a single starting configuration for the agents during training, and different starting configurations during testing. While I agree with this to some extent, I also think this argument may not be fully right. When the probing agent is testing the expert, it is essentially showing the imitator many different configurations of the environment. It may not be that it changes in the first time step (for obvious reasons), but it is essentially showing it many configurations of the expert. A more drastic change of the environment could make for a stronger argument.


4) Conclusion:
Overall, I like the idea of having a policy that tries to figure out the general behavior of a demonstrator by probing it. Having said that, I feel this paper needs to improve in the aspects mentioned above. If the authors present more convincing evidence that successfully address the comments above, I am willing to increase my score.

---

> ### Author Response · Authors · 2018-11-25
> **Author response**
>
> Thank you for your detailed reviews and constructive suggestions. We have added the suggested baselines in the revision. Here are our responses to your questions and comments:
>
> 1. Equation 1 typo?
> It is not a typo. Our reward function is different from existing curiosity reward. We are using the change of the real time m^t and m^{t-1} as the reward for inciting behavioral change from the demonstrator. We have shown more analysis and visualization to explain why this works in the new revision (Appendix B.2 & B.3). Our “self-supervised” baseline is actually using the prediction loss as reward, and it has a worse performance compared to ours.
>
> 2. Baseline missing: Random actions from expert
> Figure 5 shows the results where 10% actions from the demonstrator are purely random. With the randomness, our approach is still be able to find meaningful probing policy. We have also evaluated the success rate when we use the policy learned from the suboptimal demonstration (10% random actions). As reported in the updated Table 1, this policy is comparable to the one learned from optimal demonstrations, and it still outperforms baselines which are all trained from optimal demonstrations.
>
> 3. Baseline missing: Simple RNN policies that communicate hidden states
> We have evaluated this baseline in the revision (i.e., the “2-LSTM” baseline). The network architecture is illustrated in Figure 16. It indeed performs much worse than our full model.
>
> 4. Ablation study for the importance of fusion
> We have added the result of this baseline (i.e., the “ours w/o fusion” baseline), where we concatenate the state feature and the latent vector m^t together. The results have validated the importance of using the attention-based fusion layer.
>
> 5. Generalizability argument
> Our main idea is to show as many configurations as possible to the learner by learning a good probing policy. Since the probing always starts from a single setting, there is indeed a limit in terms of how different the new settings could be. E.g., in Maze Navigation, it is impossible for the learner to change the room layout drastically in the time limit, so the learned policy won’t make sense in a very different room layout (e.g., 8 rooms instead of 4 rooms). To obtain a better generalization, we may need to use a better imitation learning approach to replace the current one (behavioral cloning), and possibly using multiple starting configurations. But we think that it is somewhat orthogonal to our main contribution. The objective of our approach is to discover more diverse settings/configurations and consequently improve whatever imitation learning approach we actually use.

---

> > ### Comment · AnonReviewer4 · 2018-12-03
> > **Thank you**
> >
> > Thanks for the clarification on equation 1. I see a clearer picture of this now. For the baseline simply using random actions instead of actions from the expert, the results show that your method indeed is better than simply using random actions. However, it would be good if a later version of the paper shows results simply using 100% of random actions (i.e. completely removing the expert). I expect this would more strongly showcase your paper to the reader. Thanks for addressing points 4 and 6 as well. I still feel the entire dataset is engineered around the method, but I have increased my score because the method itself is interesting.

---

### Author Response · Authors · 2018-11-25
**Submission Revision**

We thank all reviewers for their constructive comments. We have added extensive new experiments, analysis, visualization and discussions in the revision as requested by the reviewers. Here is a brief summary:
1) New baselines: i) “2-LSTM” where we use two LSTMs for the demonstrator’s policy and the probing policy (its architecture is illustrated in Figure 16), and ii) “ours w/o fusion” where we replace the fusion layer with concatenation of state feature and latent vector m^{t-1}. Please see the updated Figure 4, Figure 5 and Table 1 for the new results.
2) Evaluated the success rate of the policy learned from suboptimal demonstrations, which is reported in Table 1.
3) Show variance from multiple runs in Figure 7, 8 & 9.
4) Visualized obtained latent vectors with or without probing in Figure 10.
5) Visualized and computed the correlation between the change in the latent vector m^t and the change in policy (Figure 11). This empirically proves that we indeed can use the distance between m^t and m^{t-1} as an indicator of policy change.
6) Added discussion on optimal teaching and active learning in IRL in Section 2.
7) Added more implementation details of the fusion layer in Appendix C.
8) Explained the implementation of the demonstrator’s policy in Appendix D.
9) Provided the exact reward functions used for baselines (i.e., “count-based” and “self-supervised”) in Appendix E.

---

### Meta-Review · Area_Chair1 · 2018-12-14

**Confidence:** 4
**Recommendation:** Reject

**Metareview:**

The submission proposes a setting of two agents, one of them probing the other (the latter being the "demonstrator"). The probing is done in a way that learns to imitate the expert's behavior, with some curiosity-driven reward that maximizes the chance that the probing agents has the expert do trajectories that the probing agent hasn't seen before.

All the reviewers found the idea and experiments interesting. The major concern is whether the setup and the environments are too contrived. At least 2 reviewers commented on the fact that the environments/dataset seemed engineered for success of the given method, which is a concern about how this method would generalize to something other than the proposed setup.

I also share the concern with R3 regarding the practicality of the proposed method: it is not obvious to me what problems this would actually be *useful* for, given that the method requires online interaction with an expert agent in order to succeed. The space of such scenarios where we can continuously probe an expert agent many many times for free/cheap is very small and frankly I'm not entirely sure why you would need to do imitation learning in that case at all (if the method was shown to work using only a state, rather than requiring a state/action pair from the expert, then maybe it'd be more useful).

It's a tough call, but despite the nice results and interesting ideas, I think the method lacks generality and practical utility/significance and thus at this point I cannot recommend acceptance in its current form.